# Can Remote Sensing Fill the United States' Monitoring Gap for Watershed Management?

**Vamsi Krishna Sridharan** [1,*], **Saurav Kumar** [2,*] and **Swetha Madhur Kumar** [3]

1   Fisheries Collaborative Program, University of California, Affiliated with Fisheries Ecology Division, Southwest Fisheries Science Center, National Marine Fisheries Service, National Oceanic and Atmospheric Administration, Santa Cruz, CA 95060, USA

2   Texas A&M Agrilife and Department of Biological and Agricultural Engineering, Texas A&M Agrilife Research Center, El Paso, TX 79927, USA

3   Client Analytics, Evolent Health, Arlington, VA 22203, USA; mk.swetha89@gmail.com

*   Correspondence: vamsi.sridharan@noaa.gov (V.K.S.); saurav@tamu.edu (S.K.)

**Abstract:** Remote sensing has been heralded as the silver bullet in water quality modeling and watershed management, and yet a quantitative mapping of where its applicability is likely and most useful has not been undertaken so far. Here, we combine geospatial models of cloud cover as a proxy for the likelihood of acquiring remote scenes and the shortest time of travel to population centers as a proxy for accessibility to ground-truth remote sensing data for water quality monitoring and produce maps of the potential of remote sensing in watershed management in the United States. We generate several maps with different cost-payoff relationships to help stakeholders plan and incentivize remote sensing-based monitoring campaigns. Additionally, we combine these remote sensing potential maps with spatial indices of population, water demand, ecosystem services, pollution risk, and monitoring coverage deficits to identify where remote sensing likely has the greatest role to play. We find that the Southwestern United States and the Central plains regions are generally suitable for remote sensing for watershed management even under the most stringent costing projections, but that the potential for using remote sensing can extend further North and East as constraints are relaxed. We also find large areas in the Southern United States and sporadic watersheds in the Northeast and Northwest seaboards and the Midwest would likely benefit most from using remote sensing for watershed monitoring. Although developed herein for watershed decision support in the United States, our approach is readily generalizable to other environmental domains and across the world.

**Keywords:** remote sensing; geographical information systems; watershed management; water quality; decision support; ambient monitoring; data collection





## 1. Introduction

Of the 37.6 million waterbodies including canals, stream segments, ponds, and lakes in the United States, fewer than three million are monitored in situ, with only about 60,000 monitoring sites providing information that can be compared with remotely sensed data (Figure 1a) [1,2]. Of the millions of waterbodies, only 430,893 have been assessed for water quality impairments as of 2022 (Figure 1b) [3,4]. Within the recent past, even for the assessed waterbodies, ambient monitoring of general state of the water quality and synoptic data collection of the load, transport, and fate processes associated with impairment happens sporadically in only about a third of the cases [5]. Remote sensing, both with airborne and space-borne platforms offers tremendous potential for bridging these massive data gaps. But there are significant data gaps to even be able to use remote sensing everywhere in the United States (Figure 1a). Our contribution herein is, albeit with several simplifying assumptions, to map out the potential for remote sensing to be a viable monitoring tool for all the subwatersheds—the smallest catchment class associated with waterbodies as defined by the United States Geological Survey (USGS)—within the conterminous United

States. As a corollary to these maps, we also present a geospatial estimate of where remote sensing is likely to have the greatest impact in the country by mapping the intersection of remote sensing potential and high risk of impairment and low data coverage. These maps will serve as decision support tools for researchers and practitioners to plan where to invest their energies and resources with respect to data collection for water quality management approaches.

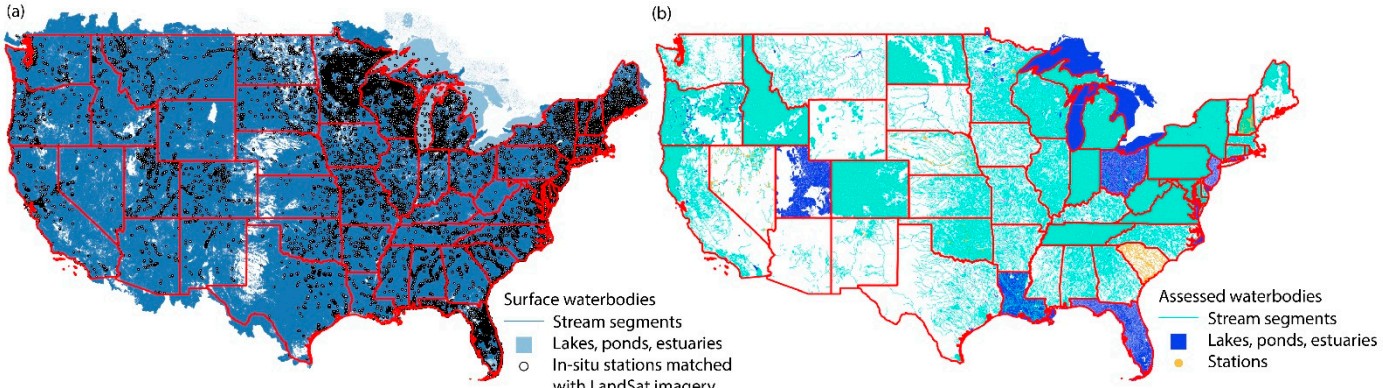

**Figure 1.** Assessment and in-situ monitoring gap for water quality in the United States: (**a**) 39.6 million stream segments, ponds, lakes and estuaries in the National Hydrography Dataset and 603,433 in-situ monitoring locations from the AquaSat database for Secchi disk depth, Chlorophyll-*a*, total suspended solids and dissolved organic carbon that have been matched to landSat scenes, and (**b**) the 430,893 assessed waterbodies in the United States obtained from the United states Environmental Protection Agency's Assessment and Total Maximum Daily Load Tracking and Implementation System (ATTAINS) database. Red lines indicate state borders.

Our focus in this paper is not to limit our treatment to a specific land surface process, water quality parameter, type of impairment or remote sensing data source. This is because water quality is a complex system response within the watershed to surface, subsurface and hydrologic processes that are modulated by human influence. Therefore, rather than focusing on a specific aspect of watershed processes, we develop a comprehensive scoping tool for adopting remote sensing in monitoring and data collection. Scoping tools such as the one presented here can be developed for specific land surface process monitoring, or water quality management of given impairments by using information relevant to certain remote sensing platforms and data collection methods.

Over the past two decades, a large number of many aerial and space-borne platforms and sensors that sample different parts of the electromagnetic spectrum have come online [6]. Platforms such as LandSat, GeoEye, WorldView, and technologies such as Light Detection and Ranging (LiDAR) are useful for water body delineation (e.g., [7]), while bathymetry of waterbodies can be obtained either from LiDaR, or by analyzing spectral band ratios of satellite and drone imagery [8,9]. Streamflow can be obtained using Radio Detection and Ranging (RaDAR) altimetry and rating curves from platforms such as Jason-3, Sentinel-3, and Saral/ALtika [10]. The terrestrial water budget can be quantified using gravimetric data, while floods can be characterized using optical sensors such as the Advanced Very-High-Resolution Radiometer (AVHRR), the Visible Infrared Imaging Radiometer Suite (VIIRS), the Moderate Resolution Imaging Spectroradiometer (MODIS), Sentinel, Landsat, Satellite Pour l'Observation de la Terre (SPOT), the Advanced Spaceborne Thermal Emission and Reflection Radiometer (ASTER), IKONOS, Worldview, RapidEye, Ziyuan 3 and Gaofen and synthetic aperture radar data [10]. Water quality parameters that are directly observable can be monitored using LandSat, Sentinel, MODIS, the MEdium Resolution Imaging Spectrometer (MERIS) and other specialized sensors such as Hyperion (chlorophyll-*a*), SPOT, AVHRR (Total suspended sediments), IKONOS (water clarity) and the Coastal Zone Color Scanner (CZCS) [colored dissolved organic matter] imagery [10].

Hyperspectral imagery [11], LiDAR and airborne laser scanning [12] can be used to infer crop characteristics and tree canopies in agrarian and forested watersheds. Within watersheds, indices developed by using reflectances from different spectral bands are useful for classifying land cover and land use types [13]. Also within watersheds, the state of best management practices such as low impact developments and green infrastructure can be monitored using visible and thermal imagery and spectral reflectance-based indices [14].

Authors have developed an array of methods ranging from the well-established to the experimental frontier have emerged to quantify watershed loading and receiving water quality pertaining to various types of impairment [15,16]. Broadly, for land surface processes such as crop cycles, land cover and land use changes, observations of surface reflectances and ratios of reflectance across multiple spectral bands are used with validation datasets such as land use classification maps or photographs are common (e.g., [17]). For water quality parameters that affect the inherent and apparent optical properties of the water column and spectral properties of the reflectances, such as total suspended solids, Chlorophyll-a, colored dissolved organic carbon, turbidity, and water surface temperature, statistical models linking observations of surface reflectances and ratios of reflectance across multiple spectral bands with the water quality parameters are popular [18]. For such processes, bio-optical and bio-geo-optical models that mechanistically link the irradiance-reflectance ratio across multiple spectral bands to the bulk water column properties are a reliable alternative [19]. For impairments such as harmful algal blooms and oxygen depletion, more sophisticated statistical (e.g., [20]) or mechanistic models (e.g., [21]) are used to relate watershed nutrient loads with impairment. For water quality parameters that do not directly affect the optical properties of the water column and spectral properties of the reflectances (e.g., mercury and heavy metals), statistical, machine learning and artificial intelligence models relating the quantities of interest with other water constituents (e.g., turbidity) are becoming increasingly common [15].

Two unifying requirements of all of these approaches are that for a given location, (i) it must be possible to obtain remotely sensed spectral imagery in the first place, and (ii) the land surface process such as fertilizer application, presence of a pollutant load attenuation best management practice like a detention pond or bioswale, and the receiving water quality must all be ground-truthed with in-situ observations [22–24]. This means that the vast potential of remote sensing can only truly be exploited in those areas where atmospheric conditions are generally conducive to remote sensing, and where some access to the watershed is possible. We leverage geospatial datasets containing information on atmospheric conditions and accessibility to rank subwatersheds according to their propensity to being amenable to remote sensing for water quality monitoring under a variety of cost-payoff scenarios.

Reasons for the gap between the number of this assessment and monitored waterbodies and the total number of waterbodies in the United states can include (i) applicability, (ii) accessibility, (iii) resource constrains and (iv) socioeconomics. First, in most cases, waterbodies might be located in remote, locations with minimal human development and limited beneficial uses to society apart from preserving pristine natural beauty and bounty. Second, there may be access issues due to rugged landforms, impassable land use features and limited mobility networks which make monitoring difficult or even impossible. Third, state, tribal and local agencies tasked with assessing and programmatically attaining designated uses for waterbodies may be hampered by resource constraints that make regular ambient monitoring and synoptic data collection difficult or even impossible in all watersheds. Fourth, there may be environmental justice issues with impairment being linked to socioeconomic and demographic conditions (e.g., [25]). In this paper, we only look at the first three factors, as we are only interested in the technical constraints of feasibility of using remote sensing for water quality monitoring and data collection here. The advantage of remote sensing over conventional monitoring programs is that for areas where the former is applicable, ground-truthing need not be performed very often, and in

the overall life cycle of monitoring and data collection programs, can be significantly less expensive to implement, particularly if public domain data and tools are used.

The paper is organized as follows: in Section 2, we describe our classification approach, the datasets we used and the geostatistical methods we adopted. In Section 3, we showcase the remote sensing potential map. In Section 4, we discuss where in the country gaps need to be filled, given the potential for remote sensing by looking at the applicability and accessibility of the subwatersheds and the intersection of quantified risks of impairment, ecological vulnerability and lack of coverage. Finally, also in Section 4, we outline the caveats in our approach, and chart our future trajectory in making these classifications more robust and readily accessible to the watershed management community.

## 2. Materials and Methods

Our fundamental premise is that we can determine the potential of remote sensing to be useful for ambient monitoring and synoptic data collection in a given waterbody by mapping this potential at the USGS' Hydrologic Unit Code 12 (HUC-12) subwatershed scale. The HUC-12 subwatershed classification allows us to analyze watersheds at the 1:24,000 scale. At this scale, these subwatersheds typically contain four stream segments, except in a few larger subwatersheds in the Northeast and the Great Lakes. We feel that this spatial scale is sufficient to identify the potential for remote sensing, as the streams encompassed within these subwatersheds are usually higher order distributaries or small areal bodies with similar climate, land use and terrain characteristics. In total, there are 102,973 HUC-12 subwatersheds in the conterminous United States. We discuss our approach in Section 2.1 below.

We used numerous Geographical Information System (GIS) resources in this study to map access, acquisition, constraints and risks of impairment. All GIS operations and visualizations were performed in QGIS [26], while the model development was performed in MATLAB [27]. In determining the potential for remote sensing and its applications, we used existing accessibility and image acquisition conditions, and did not account for the effects of land use and land cover change or climate change. After determining the potential of remote sensing as a data source within each subwatershed, we subsequently subset subwatersheds on the basis of applicability and resource constraints as described in Section 2.2 below. We have listed all the datasets we used in these analyses in Table 2.

**Table 1.** Data sources used in this study.

| Serial Number | Data | Use | Source | Location |
|---|---|---|---|---|
| 1 | Hydrologic Unit Class 12 subwatersheds | Smallest hydrologic feature for analysis | National Hydrography Dataset | https://www.usgs.gov/national-hydrography/national-hydrography-dataset (accessed on 20 April 2022) |
| 2 | Digital Elevation Model | Hillshade and basemaps | Advanced Spaceborne Thermal Emission and Reflection Radiometer (ASTER) | https://earthexplorer.usgs.gov (accessed on 20 April 2022) |
| 3 | Cloud cover database | Acquisition analysis | EarthEnv | https://www.earthenv.org/cloud (accessed on 20 April 2022) |
| 4 | Time to cities | Accessibility analysis | Malaria Atlas Project | https://malariaatlas.org/research-project/accessibility-to-cities/ (accessed on 20 April 2022) |
| 5 | United States administrative boundaries | Basemaps | Census cartographic boundary files | https://www.census.gov/geographies/mapping-files/time-series/geo/cartographic-boundary.html (accessed on 20 April 2022) |

**Table 2.** *Cont.*

| Serial Number | Data | Use | Source | Location |
|---|---|---|---|---|
| 6 | Assessed waterbodies | Evaluating assessment gaps | Assessment and Total Maximum Daily Load Tracking and Implementation System (ATTAINS) | https://www.epa.gov/waterdata/attains (accessed on 20 April 2022) |
| 7 | Population by HUC-12 | Evaluating human footprint | EnviroAtlas | https://www.epa.gov/enviroatlas (accessed on 20 April 2022) |
| 8 | Agricultural, domestic, industrial and thermoelectric water demand by HUC-12 (MGD) | Evaluating water supply needs | EnviroAtlas | https://www.epa.gov/enviroatlas (accessed on 20 April 2022) |
| 9 | Protected areas under the International Union for Conservation of Nature and the United States protected areas by HUC-12 (% of land cover) | Evaluating conservation need | EnviroAtlas | https://www.epa.gov/enviroatlas (accessed on 20 April 2022) |
| 10 | Big game and bird hunting, fishing, and migratory bird watching demand by HUC-12 (days/year) | Evaluating recreational need | EnviroAtlas | https://www.epa.gov/enviroatlas (accessed on 20 April 2022) |
| 11 | Vulnerability index of native aquatic species by HUC-12 | Evaluating biodiversity needs | EnviroAtlas | https://www.epa.gov/enviroatlas (accessed on 20 April 2022) |
| 12 | Wastewater discharge (MGD), total permitted discharge, daily agricultural runoff (mm) | Evaluating environmental needs | EnviroAtlas | https://www.epa.gov/enviroatlas (accessed on 20 April 2022) |

Although our analysis pipeline extends to Alaska, Hawaii and other minor islands and United States protectorates, we have restricted all figures to only the conterminous United States for simplicity. To illustrate the role of topography on various watershed characteristics, in most figures, we have superimposed our analytical products on a hillshade relief we generated from the 30 m resolution ASTER digital elevation model obtained from the USGS Earth Explorer portal.

*2.1. Potential for Remote Sensing as a Tool to Collect Data on the Subwatershed Scale*

We used the mean annual fraction of cloudy days from the EarthEnv global cloud cover model dataset [28] as a surrogate to indicate how likely it is to acquire remote sensing imagery of a 1 Km$^2$ scene and how hard it is to process this imagery for use (Figure 2a). This dataset includes 1 Km resolution mean conditions between the period of 2001 and 2014. We used the global map of minimum travel times from a location to a population center (defined as a city or township with more than 50,000 people) produced by the Malaria Atlas Project [29] as a surrogate for accessibility to a given 1 Km$^2$ grid cell (Figure 2b). This raster map is the outcome of a geospatial travel time model that takes existing road and rail networks, land cover and 30 m resolution digital terrain models of the entire globe. The combination of these two datasets, respectively indicate how likely it is to acquire a scene remotely, and how easy it is to ground-truth reflectances to link the remotely sensed data to land cover characteristics and water quality impairments. The tacit assumption is that it will be possible for stakeholders tasked with managing the watershed to reach any major population center, and then travel from there to perform in-situ sampling work. To link these datasets to the hydrological units of interest, we used the USGS' National Hydrography Dataset's (NHDplus) HUC-12 layer [3].

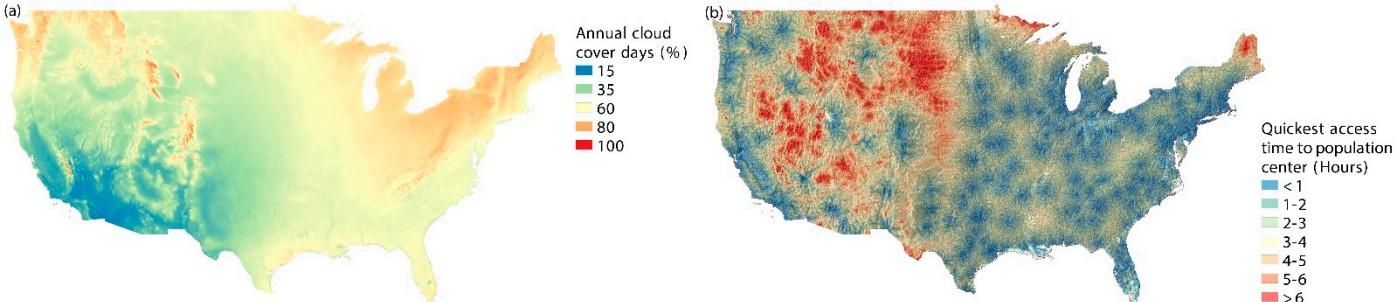

**Figure 2.** Datasets used to model potential for remote sensing in watershed monitoring: (**a**) mean annual fraction of cloudy days, and (**b**) maximum shortest travel time to population center within each HUC-12 subwatershed. These quantities are surrogates for acquisition and accessibility of remotely sensed scenes and in-situ data needed to ground-truth them.

We modeled the potential for using remote sensing for monitoring and data collection within each subwatershed as the product of various categorical levels of $\tau_{\text{max}}^{\text{Shortest}}$, the maximum of the shortest time to reach a population center from anywhere within the subwatershed, and various categorical levels of $\mu_{\text{Cloud Cover}}$, the mean annual fraction of cloudy days within the subwatershed. Within each subwatershed, we estimated $\tau_{\text{max}}^{\text{Shortest}}$ as the maximum over all pixels within the area of the shortest time to reach a population center. We estimated $\mu_{\text{Cloud Cover}}$ as the mean over all pixels within the area of mean annual fraction of cloudy days. Thus, for the $i$th subwatershed, we have

$$P_{\text{Remote Sensing},i} = \mathbb{C}\left(\tau_{\text{max},i}^{\text{Shortest}}\right) \times \mathbb{C}(\mu_{\text{Cloud Cover},i}) \tag{1}$$

Here, the function $\mathbb{C}(\cdot)$ assigns a category from 0 to 3 to both the acquisition ($\mu_{\text{Cloud Cover}}$) and access ($\tau_{\text{max}}^{\text{Shortest}}$) variables according to Table 3. 0 indicates a low potential for success, and 3 indicates the highest potential for success. The possible values of $P_{\text{Remote Sensing}}$ obtained in this way are binned into four categories for ease of application: 0 (unsuitable), 1–3 (low), 4–6 (good), and 9 (excellent). We note that the numeric values of $P_{\text{Remote Sensing}}$ themselves are only meaningful when they contribute to the different categories of "unsuitable," "low," "good" and "excellent."

**Table 3.** Cost-payoff matrix for modeling potential of remote sensing for watershed monitoring.

| Category | Accessibility: Shortest Time to Population Center | Acquisition: Fraction Cloudy Days Per Year |
|---|---|---|
| Conservative | | |
| 0 | >6 h | >75% |
| 1 | 2–6 h | 50–75% |
| 2 | 1–2 h | 25–50% |
| 3 | <1 h | <25% |
| Normal | | |
| 0 | >24 h | >90% |
| 1 | 6–24 h | 75–90% |
| 2 | 3–6 h | 50–75% |
| 3 | <3 h | <50% |
| Optimistic | | |
| 0 | >48 h | >95% |
| 1 | 24–48 h | 90–95% |
| 2 | 6–24 h | 75–90% |
| 3 | <6 h | <75% |

As stakeholders in the watershed management domain may weigh the costs associated with the logistics of travel to remote locations for in-situ ground-truthing of remote sensing data, and the careful planning of periods when to collect in-situ data in conjunction with clear sky days, we represented these weights in cost-payoff matrices to set the categorical

values of the acquisition and access variables (Table 3). These weights effectively lower the $\tau_{\text{max}}^{\text{Shortest}}$ and $\mu_{\text{Cloud Cover}}$ threshold values, respectively for the accessibility and acquisition criteria for the different categories going from conservative to normal to optimistic estimates of the cost-payoff relationships (Table 3).

For example, one regulatory agency may be willing to send an engineer on a six-hour long journey to collect water quality data. So, any value of $\tau_{\text{max},i}^{\text{Shortest}}$ smaller than six hours for a subwatershed within the jurisdiction of this agency would get a $\mathbb{C}(\tau_{\text{max}}^{\text{Shortest}})$ value of 3. Another agency may decide that it does not want its engineers to spend more than one hour traveling to a remote site. In this case, the $\mathbb{C}(\tau_{\text{max}}^{\text{Shortest}})$ for any $\tau_{\text{max}}^{\text{Shortest}}$ larger than one hour would be 2 or lower. Similarly, a consultant may be willing to invest the effort into building robust processing pipelines to deal with largely cloudy scenes or invest time and resources in data fusion-based cloud removal algorithms that are becoming more popular across various platforms [30,31]. This consultant would perhaps be willing to assign a $\mathbb{C}(\mu_{\text{Cloud Cover}})$ value of 2 or higher for a subwatershed where the sky is likely to be cloudy for almost 75% of the days in the year. Another consultant who is perhaps geared to directly use analysis-ready products may assign a $\mathbb{C}(\mu_{\text{Cloud Cover}})$ value of 3 for their subwatershed only if the sky is likely to be clear for 75% of the days in the year. By presenting maps of the categorical levels of the potential across a range of $\mathbb{C}(\cdot)$ values, we allow decision-makers to choose the cost-payoff level they are comfortable with to plan their workflows accordingly.

### 2.2. Quantifying Benefits of Remote Sensing in Watershed Monitoring

We modeled the potential for remote sensing using Equation (1) and surrogates for accessibility and acquisition. However, these maps in themselves provide only one piece of the puzzle. It is only when the potential for watershed monitoring using remote sensing is combined with risk maps and lack of coverage by conventional monitoring that areas can be identified where remote sensing can play a crucial role. To do this, we developed maps of five metrics of risk using published GIS data from the EnviroAtlas [32] (Table 2): (i) the human footprint, (ii) anthropogenic water demand, (iii) ecosystem vulnerability, (iv) impairment, and (v) conventional monitoring and assessment coverage gaps. As the units of these metrics are different from one another, we simply overlaid these maps and determined the intersecting areas to determine where the remote sensing role would be most useful.

*Human footprint*: We collected the human population in each subwatershed. The conventional wisdom is that impairment and watershed management matter most where human settlements are prevalent. The subwatersheds populations were then ranked from 0 to 6 with decadal increase from less than 10 to more than one million people.

*Anthropogenic water demand*: We combined the domestic, agricultural, industrial and thermoelectric water demand in each subwatershed to estimate the total water demand from each subwatershed. Watersheds with greater demand are likely to experience larger ecological deficit flows and a higher propensity for critical conditions of impairment. The subwatershed demands were then ranked from 0 to 6 with decadal increase from less than 12,000 Gallons Per Day (GPD) [the per capital daily water demand is 1200 GPD estimated by dividing the total water demand by the total population].

*Ecosystem vulnerability*: We combined three key ecosystem services with equal weightage to each to estimate an overall vulnerability index. First, we summed the average number of days in a year that hunting, fishing, and recreational activities are typically engaged in within a subwatershed to obtain a fractional recreation time, $f_{\text{Recreation}}$. Large values of this number indicate greater pressure for recreational activities. Second, we combined the total fraction of land cover earmarked for conservation by both the International Union for Nature Conservation (IUCN) and the United States Government, $f_{\text{Conservation}}$. Large values of this number indicate that additional monitoring, assessment, and restoration effort must be expended in these subwatersheds. Third, we collected the native aquatic species vulnerability index, $f_{\text{Vulnerability}}$, a key measure of how many endangered, native

species are threatened [33]. This index is also a proxy for ecosystem health in general. Higher values of this index imply generally poorer ecosystem health. All these numbers range from 0 to 1. The overall ecosystem vulnerability of the subwatershed is then

$$f = w\Big( f_{\text{Recreation}} \mathbb{I}_{\text{Recreation}} + f_{\text{Conservation}} \mathbb{I}_{\text{Conservation}} + f_{\text{Vulnerability}} \mathbb{I}_{\text{Vulnerability}} \Big) \qquad (2)$$

where $\mathbb{I}_i$ is an indicator that can either be 0 or 1 depending on whether the service $i$, i.e., recreation, conservation or species vulnerability, is being provided by the subwatershed or not, and $w$ is 1, 0.5 or 0.33 if there are one, two or all three of the services being provided by the subwatershed. The subwatersheds ecosystem vulnerability indicator was then ranked 0 for no vulnerability, and then from 1 to 4 in increments of 0.25.

*Impairment risk*: We developed subindex curves of the total wastewater discharge in Million Gallons per Day (MGD), the total permitted pollutant load in pounds per year, and the total agricultural overland, tile and non-tile subsurface runoff in mm as described in Walsh and Wheeler [34]. We then combined these subindices using a maximum subindex measure as the overall index of impairment within a subwatershed. This index was deemed to represent the impairment risk in the subwatershed. To develop these subindex curves the following approach was adopted: first, we summed the total load in each class, i.e., wastewater, permitted sources, and agricultural runoff, and divided it by the total surface area of waterbodies within the watershed to normalize loads across subwatersheds. Then we omitted zero values and obtained the 25th, 50th and 75th quantiles of these normalized loads across all the subwatersheds in the country and assigned rank values from 0 to 4 depending on whether there was no impairment from that loading class to whatever quantile range the normalized loads fell into. This effectively positioned the normalized loads in each load class onto a subindex curve. Then, as the nature of impairment is likely to vary depending on the dominant economic sector, human activity, and land surface processes in each subwatershed, we took the overall impairment risk to be the maximum of the three rank values. Thus, higher values of this index (ranging from 0 to 4) represents higher risk of impairment.

*Conventional monitoring and assessment coverage gaps*: For each subwatershed, we estimated the total length of assessed stream segments, $L_{\text{Stream}}$, the total area of assessed ponds, lakes, and estuaries, $A_{\text{Body}}$, and the total number of monitoring stations, $N_{\text{Station}}$, from the Assessment and Total Maximum Daily Load Tracking and Implementation System (ATTAINS) database [4]. A subwatershed was determined to have a monitoring gap if $L_{\text{Stream}} + A_{\text{Body}} + N_{\text{Station}} = 0$.

Finally, we also produced a remote sensing monitoring potential map using the normal cost-payoff estimates from Table 3 to indicate where remote sensing would be likely feasible. Subwatersheds where the role of remote sensing is likely to be most crucial were then classified as those that met the decision tree in Figure 3, that is, where all the layers after suitable thresholding to represent various risks intersect with at least good potential for remote sensing.

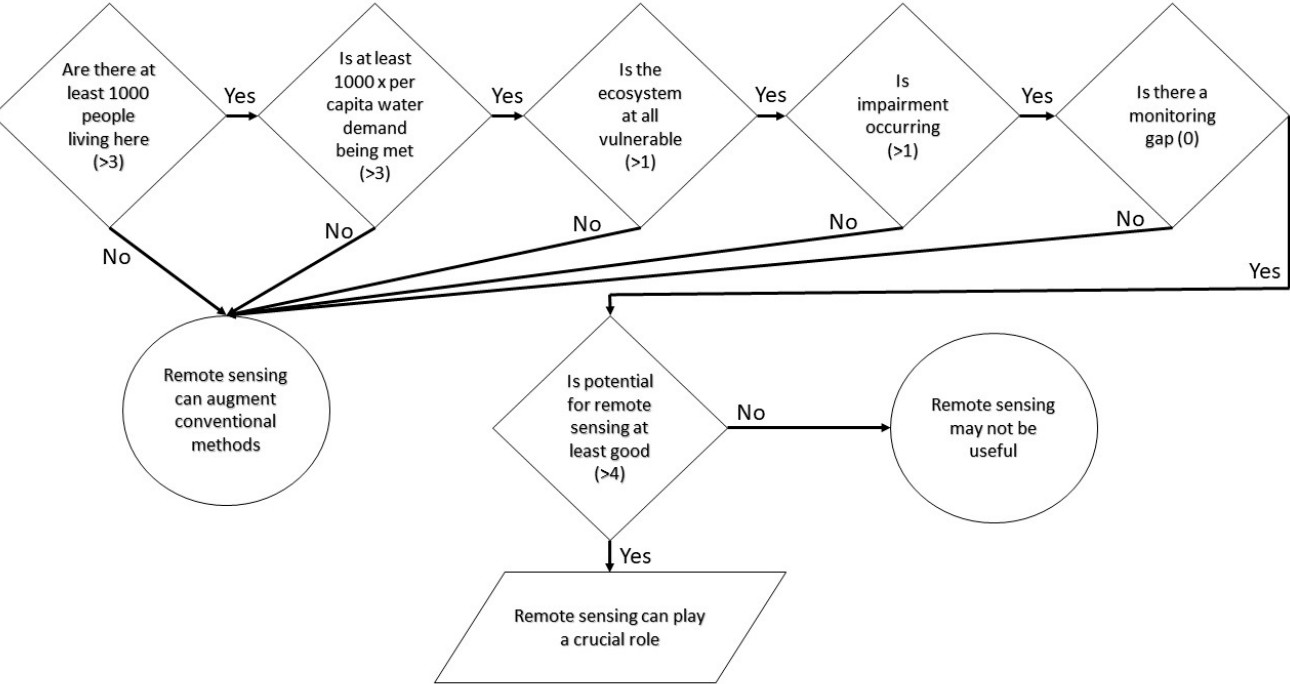

**Figure 3.** Flowchart for estimating where the role of remote sensing will be crucial in watershed management. The operation represents the intersection of geospatial layers. The numbers indicate threshold values of various index levels.

## 3. Results

We show the maps of potential for remote sensing in monitoring within watersheds for various cost-payoff scenarios in Figure 4. These potential rankings represent an interplay between the ambient cloud cover (Figure 2a) and the maximum of the shortest travel time to a population center (Figure 2b). The model predicts that when cloud cover is low, and travel times are short, the potential is maximized. Conversely, when cloud cover is high and travel times are long, the potential is inhibited. Typically, cloud cover is lowest in the Southwest, and increases generally through the Central Plains towards the East and is highest over the North and the Rockies and Appalachian mountains (Figure 2a). Travel times are generally relatively low (less than three hours) on the Eastern and Western seaboards, and throughout the Central plains and in the South, and increase to more than six hours in the mountainous Rockies and the remote Midwest. These patterns are reflected in the model.

Going from left to right in Figure 4, the expense of effort to build pipelines for occluded images and changing light conditions is budgeted increasingly generously. This reflects in the improved potential for remote sensing in more parts of the Mideast and East. Going from top to bottom in Figure 4, the effort required to collect in-situ data for ground-truthing is budgeted increasingly generously. This reflects in the improved potential for remote sensing in more parts of the Southwest and the Midwest, until in the bottom right of Figure 4, there is more or less uniformly high potential throughout the country. Nonetheless, most of the California Central Valley, Southern California, and the Central Plains have uniformly good to excellent potential under all cost-payoff scenarios, owing to generally clear skies and short travel times.

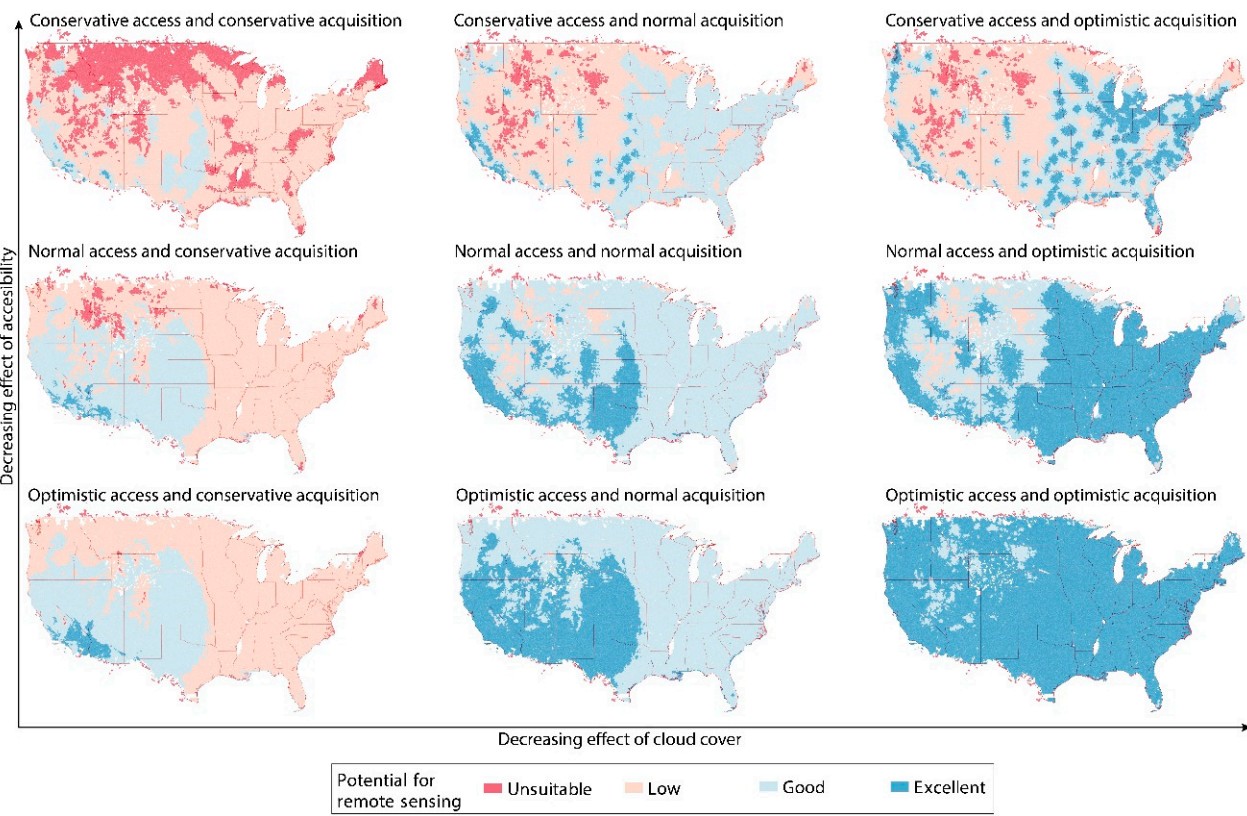

**Figure 4.** Modeled maps of the potential of remote sensing for data collection and monitoring of watersheds under various cost-payoff assumptions. As the effect of cloud cover decreases (more cloudy days are acceptable), more cloudy areas display higher potential. As the effect of access decreases (longer travel times are acceptable), more remote areas display higher potential. Thin red lines are state boundaries.

In Figure 5, we show the various risk metrics (a through e) and the associated remote sensing utility map (f) by applying the flowchart in Figure 3 to each subwatershed in these layers. Population centers, as expected, are generally concentrated near the major cities on the Eastern and Western seaboards and in the Mideast and the Gulf of Mexico coast (Figure 5a). The water demand generally tracks the population centers, except in California, the Midwest, and the Central Plains subwatersheds from where water is diverted to other places (Figure 5b). For instance, in California, water is gravity fed from reservoirs in the North to consumers in the South. Almost everywhere in the United States, ecosystems are vulnerable (Figure 5c). In most of the Midwest, this is likely due to large tracts of lands being designated as protected zones, and in the Sierra Nevada mountains in the West, due to ecological disasters by sustained drought and forest fires. Most of the population centers on the Eastern and Western seaboard are polluted by wastewater and permitted discharges, while the Central plains and Midwest impairments reflect of agricultural runoff into the Mississippi River tributaries (Figure 5d). The ATTAINS database indicates that most of the nation's subwatersheds have at least some waterbodies that have been assessed. But many subwatersheds in the United States-Mexico border region, the remote Midwest, and the Rockies remain unassessed (Figure 5e). Based on a combination of these layers in conjunction with the remote sensing potential model under the assumption of a normal cost-payoff scenario (middle panel in Figure 4 and Table 3), several hotspots emerge banded predominantly in the South and the North where remote sensing could be crucial, and often perhaps the only source of assessment of the condition of these subwatersheds (Figure 5f).

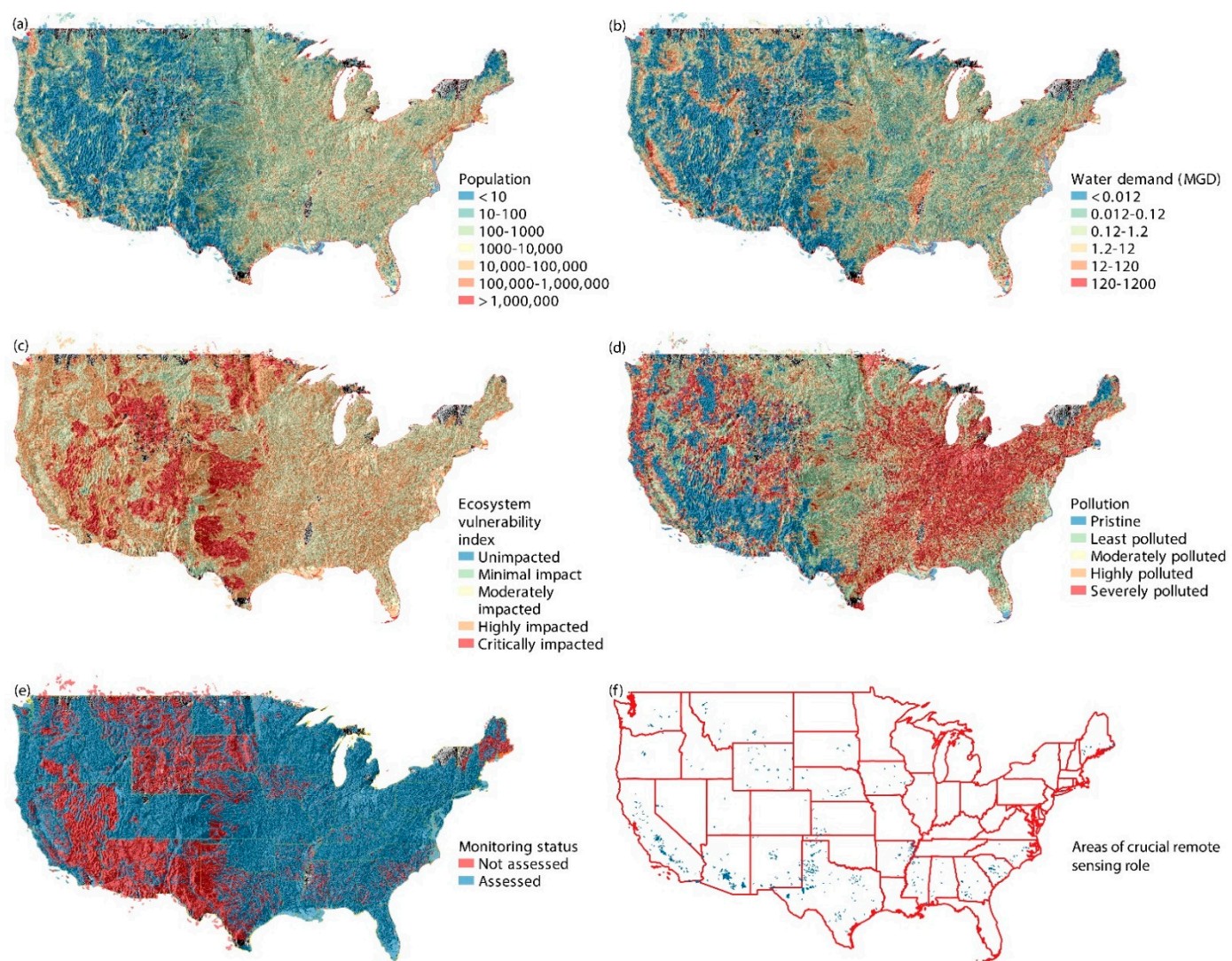

**Figure 5.** Predicted subwatersheds where the role of remote sensing could be crucial as the intersection of greater than good remote sensing potential, high impairment risk indicators, and low coverage. Across subwatersheds, (**a**) population, (**b**) total water demand, (**c**) ecosystem vulnerability index, (**d**) pollution entering streams and waterbodies, (**e**) water quality monitoring gaps, and (**f**) intersection of these layers with normal accessibility and normal cloud cover cost-payoff model predicted higher than good remote sensing potential subwatersheds (blue polygons).

It is evident from these remote sensing potential maps and the combination of risk metrics and remote sensing potential that there are specific areas within the country where watershed researchers and managers can beneficially leverage remote sensing data at various spatial scales ranging from stream segment-reach to the subwatershed scale. By allocating resources preferentially to occluded and poorly-lit scene imagery usage pipelines, the potential for using remote sensing can be tremendously maximized. This is borne out by rapidly improving potential going left to right in Figure 4, than by going from top to bottom in Figure 4.

## 4. Discussion and Conclusions

We have presented a set of maps that chart the potential of remote sensing to augment the monitoring and data collection in watersheds for land surface and water quality processes. These are available for open source download in a Git repo [35]. To enable easy access to, and visualization of these maps, the risk factors for impairment, and summary statistics of the potential of remote sensing under various assumptions within the cost-

payoff matrix, we have also developed an app [36] using Google Earth Engine [37]. This app contains an interactive map within which the user can click on a specific subwatershed of interest to obtain additional information (Figure 6). A summarization of this paper is available as a guide to using the app in a separate panel. Once a subwatershed has been clicked, a summary visual report will be generated, and the relevant information on the impairment risk factors and the potential for remote sensing can be downloaded as comma-separated value files.

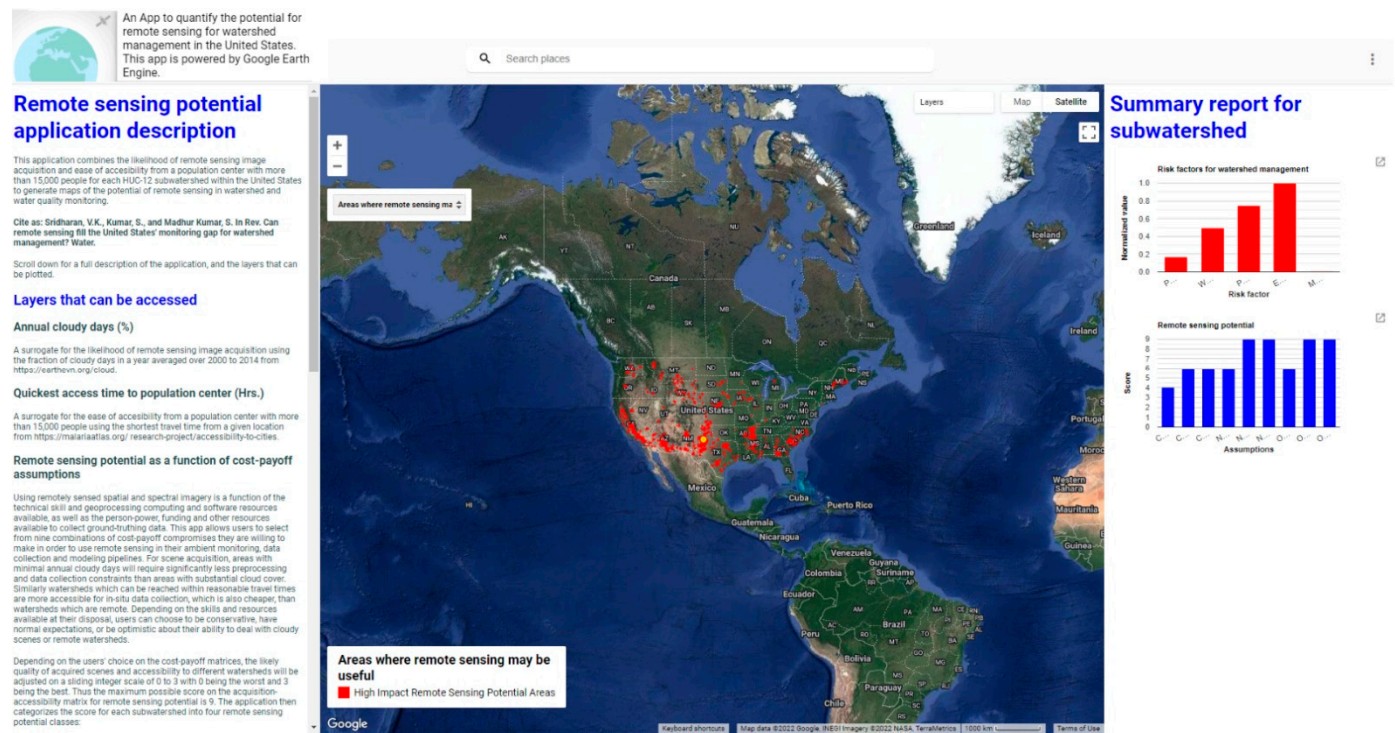

**Figure 6.** Google Earth Engine application to demonstrate the potential of remote sensing in monitoring watershed management in the United States. Attribution: Map Data © 2022 Google INEGI Imagery © 2022 NASA, TerraMetrics.

Additionally, we have shown how such maps can be modified within a cost-payoff landscape to be truly customizable by watershed decision-makers. The cost-payoff analysis also shows that improving the data processing pipelines for occluded and poorly lit scenes can tremendously benefit the power of remote sensing (going left to right in Figure 4). By combining the regions of high remote sensing potential with other risk factors describing human footprint, impairment risk, ecosystem vulnerability, and conventional monitoring coverage gaps, we can identify those subwatersheds where remote sensing is likely to have the highest benefit, and in fact, likely be an integral source of primary data. Large areas in the Southern United States, particularly in California, New Mexico, Texas, Mississippi, Georgia, and the Carolinas, and sporadic watersheds in the Northeast and Northwest seaboards (Washington and Maine) and the Midwest would likely benefit most from using remote sensing for watershed monitoring (Figure 5f).

Our approach represents a seminal and necessary step in aiding the decision-making process for resource-constrained regulatory agencies and contractors. By augmenting these maps with other socio-economic geospatial data, deeper insights can be gained to tackle challenges of environmental justice and equity. Rather than focusing on specific watershed processes, impairments, and remote sensing platforms, we have developed a method of assessing the general potential for remote sensing for whatever applications are envisioned by system managers in their monitoring and data collection workflows. This approach allows us to explore the benefits of remote sensing further, as (i) not all parts of the country

are monitored regularly, and even when they are, in-situ monitoring costs can typically exceed several hundred thousands of dollars a year within a small watershed [38], and (ii) remote sensing can provide long-term cost savings. Remote sensing potential maps could be constructed for specific applications outlined in the introduction by suitably modifying the factors considered in the model presented in Section 2.1. For example, if a consultant proposes a data fusion pipeline using Sentinel 2, MODIS, and Planet imagery [39] to study watershed health, heavy metal concentrations in a nearby lake, and the role of watershed best management practices, then the principal costs of image acquisition will likely be the cost of procuring high-resolution Planet imagery, while in-situ monitoring costs will likely be driven by personnel costs rather than access time.

There are several caveats to the maps we have produced here. First, the assumptions built into the EarthEnv cloud cover and the Malaria Atlas travel time databases carry over to our maps. While these data sources are excellent for global coverage at the kilometer-scale, our pipeline may be refined with source DEMs and spectral imagery for specific watersheds to systematically downscale our national coverage maps. The shortest time to reach a population center, while somewhat ad-hoc, is an appropriate metric to characterize the difficulty of ground-truthing remote sensing imagery because typically waterbodies outside major population centers are not monitored. However, as we show in Figure 5d, such areas are also at high risk of impairment. Second, our remote sensing potential model is extremely simple and steady state, and yet captures the underlying tradeoff between scene acquisition and ground-truthing. Within a watershed, multiple process-scales exist, ranging from storms and coastal upwelling events that happen on hourly to daily timescales, land use and land cover changes that happen on annual to decadal timescales, and landform changes that happen over many decades to centuries [40]. Our maps do not factor in these different timescales. More sophisticated models, including spatial kriging approaches and Gaussian process models that incorporate seasonal variability in scene acquisition and travel time can be developed. Cloud cover, for instance, can vary seasonally so that even in areas with many cloudy days, there can be periods with clear skies during which period field campaigns could be planned. We do not capture such nuances here. For simple applications that require only a single remote sensing image such as a National Agricultural Image Program [41] or LandSat scene, cloud cover may not even be an issue if the acquisition window can be carefully determined. Although many non-dimensional metrics for land use and land cover classification for characterizing the environment can be derived from multi- and hyperspectral imagery in most locations within the United States, as we showed in Figure 1, there are still significant data gaps that require physical ground-truthing for which our maps will be useful. Third, we have operated at the HUC-12 subwatershed scale in this project for computational tractability. However, much finer reach-scale resolution of the impaired waterbodies can be incorporated within our pipeline by combining the NHDplus and ATTAINS datasets using cloud platforms such as Google's Earth Engine [37]. Fourth, our thresholds for cloud cover and minimum travel time to population centers for the various cost-payoff scenarios are somewhat arbitrary, but still indicate sensible and meaningful trends in the remote sensing potential maps. This is a reflection of the fact that we have sacrificed specificity to different remote sensing technologies for the generality of our guidance maps. These thresholds can be more rigorously defined for different types of remote sensing platforms by combining the atmospheric conditions and ground-truthing needs of specific technologies. Fifth, while we have chosen risk metrics that largely encompass most risk factors for watershed impairments, a more thorough exploration of available datasets, particularly socioeconomic data, is possible to identify other, transdisciplinary benefits of remote sensing within a watershed. It is likely that such exhaustive inclusion must be done on a local municipal, county, or state scale than on the national scale for disparate factors to sensibly be combined at meaningful physical scales. In subsequent iterations of these products, we will tackle these advances. Sixth, we only considered terrestrial subwatersheds in our current pipeline, and excluded all coastal and

Great Lakes impairments. In subsequent iterations of these maps, we will include these areas as well.

While we have developed the pipeline for the United States, a similar workflow could be developed anywhere in the world, or even for the entire planet. The only consideration is that while Earth Explorer, the United States Environmental Protection Agency, and the EnviroAtlas catalog a wide array of geospatial datasets for immediate use, such data may be difficult to find in the international landscape, and may have to be stitched together from disparate sources of information. Nonetheless, for any serious, reproducible, and scalable application of remote sensing for monitoring and data collection on watershed loads and impairments, this is a necessary first step. For the first time ever, a product is available that will allow researchers and managers to evaluate the potential of remote sensing to augment, or even fulfill ambient monitoring and data collection needs throughout the United States. We hope that our maps will be useful for the watershed managers within the United States, even as we continue to improve them with finer granularity and additional factors.

**Author Contributions:** Conceptualization, V.K.S., S.K. and S.M.K.; methodology, V.K.S. and S.K.; software, V.K.S. and S.K.; validation, V.K.S. and S.K.; formal analysis, V.K.S.; investigation, V.K.S.; resources, V.K.S., S.K. and S.M.K.; data curation, V.K.S.; writing—original draft preparation, V.K.S.; writing—review and editing, V.K.S. and S.K.; visualization, V.K.S.; supervision, V.K.S. and S.K.; project administration, V.K.S.; funding acquisition, Unfunded study. All authors have read and agreed to the published version of the manuscript.

**Funding:** This research received no external funding.

**Institutional Review Board Statement:** Institutional review and approval was not applicable for this study.

**Informed Consent Statement:** Informed consent was not applicable for this study as there were no human subjects.

**Data Availability Statement:** Data available in a publicly accessible repository that does not issue DOIs here: https://github.com/vamsiks2003/remoteSensingPotentialMapping (accessed on 20 April 2022). Publicly available datasets were analyzed in this study. This data can be found here: https://www.usgs.gov/national-hydrography/national-hydrography-dataset, https://earthexplorer.usgs.gov, https://www.earthenv.org/cloud, https://malariaatlas.org/research-project/accessibility-to-cities/, https://www.census.gov/geographies/mapping-files/time-series/geo/cartographic-boundary.html, https://www.epa.gov/waterdata/attains, https://www.epa.gov/enviroatlas (accessed on 20 April 2022). Data citation: Sridharan, V.K., Kumar, S.N., and Madhur Kumar, S. 2021. GitHub Repo remoteSensingPotentialMapping (https://github.com/vamsiks2003/remoteSensingPotentialMapping (accessed on 20 April 2022)).

**Acknowledgments:** Discussions with Rocky Talchabadel and Santosh Palmate at Texas A&M University were useful in the development of the methods presented here. Critical review by Lee Harrison at the University of California, San Diego and recommendations by three anonymous reviewers and the Academic Editor at Water significantly improved the quality of this manuscript. The lead author acknowledges the Environmental and Water Resources Institutes Remote Sensing Task Committee, Total Maximum Daily Load Analysis and Modeling Task Committee and the Watershed Management Technical Committee whose deliberations motivated this study.

**Conflicts of Interest:** The authors declare no conflict of interest.

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
