# Peer review of "Can Remote Sensing Fill the United States’ Monitoring Gap for Watershed Management?"

_water, doi:10.3390/w14131985_

Round 1

Reviewer 1 Report

I think "Can remote sensing fill the United States’ monitoring gap for 
watershed management?" is  very interesting manuscript and has sufficient novelty. The authors just need to work more with the coherence of the presentation and also some revisions in the conclusion and references.

Author Response

We thank the reviewer for their positive feedback. We have included a section in the introduction that more thoroughly fleshes out the various types of monitoring gaps that can be filled by remote sensing for a variety of watershed and water quality processes between lines 60 and 112. We have revised Figure 1 to include a more extensive discussion of the data gaps that can be filled by remote sensing. In the discussion, we have fleshed out the various assumptions and caveats in the model between lines 411 and 475. We hope that these additions will add coherence to the paper. We have added 24 references which allow us to delve more deeply into the role of remote sensing in watershed and water quality monitoring.

Reviewer 2 Report

This manuscript deals with an important issue in applications of remote sensing from a macroscopic view, the results are quite useful and important for decision maker in the watershed management in the US. Some comments and suggestions are as follows.

l  The academic gap is clearly described in the introduction; however, the literature review is somewhat simple and should be enriched.

l  In figure 1, the legend of purple color has not provided.

l  Line 76, please check the gramma.

l  Line 105, please list the names of the four stream segments

l  In Table 1, please give the full name of “S.No.”.

l  timeto in the legend of Figure 2(b) should be “time to”.

l  Line 165:  7 and 8 are not used in your approach?

l  Figure 5 (f) is difficult to read, because there is not any legends about the colors or lines in the map.

Author Response

We thank the reviewer for the constructive feedback and recommendations.

We have performed a more comprehensive literature review and included 24 additional references which flesh out the role of remote sensing in watershed and water quality monitoring.

Figure 1 has been revised and legends are now more clearly provided. The red or purple lines indicate state boundaries.

The grammar on line 130 (previously 76) has now been corrected.

On line 102 (previously 105), we merely indicate that there are some subwatershed which contain up to four stream segments. We do not name such watersheds or stream segments within them explicitly.

We have expanded this as "Serial Number."

The legend in Figure 2(b) has been corrected.

In line 216 (previously 165), the numbers 7 and 8 do not feature. This is because ours is a multiplicative model, and so when we multiply the two factor levels of 0, 1, 2 and 3 with each other, we don't get 7 or 8. We have added an explanation on lines 217 and 218.

We have included a note in the caption of Figure 5 to make 5f more clear. The lines are state boundaries.

Reviewer 3 Report

This paper argues that remote sensing plays a major role in watershed management in the United States. It is understandable to use remote sensing data in wide area watershed management. However, I do not understand the focus of the discussion in this paper. For example, if water quality is to be monitored, more specifically, which remote sensing data (such as water quality data obtained from MODIS or Landsat) will be used and how much it will contribute to watershed management compared to the past. I think we should discuss it in detail. Unfortunately, this paper is very comprehensive and it is difficult to judge the validity of remote sensing data.

Author Response

We thank the reviewer for their comments. We agree with the reviewer that the paper is comprehensive in its approach. This is our intention, as we did not want to restrict the application to a specific type of impairment or a specific remote sensing data source. We have added significant discussion on the different types of watershed and water quality data that is available and the various remote sensing platforms and data approaches between lines 69 and 112, to clarify that our scope is comprehensive, but that we understand that there are likely to be specific applications. Between lines 60 and 68, we justify this approach. We have additionally revised Figure 1 to more explicitly motivate the need for this paper in the context of monitoring data gaps. Between lines 413 and 477, we have discussed the limitations of the model, as well as provided an example approach for considering specific datasets for a particular impairment. 

Round 2

Reviewer 3 Report

This paper is holistic even after review, and it is difficult to judge the validity of remote sensing data, but respecting the author's revisions and the opinions of other reviewers. I agree to publish if the opinions of other reviewers have been amended.